# *N*-Organothio β-Lactams Offer New Opportunities for Controlling Pathogenic Bacteria

**DOI:** 10.3390/pathogens14070628

**Published:** 2025-06-24

**Authors:** Edward Turos

**Affiliations:** Department of Chemistry, University of South Florida, 4202 East Fowler Avenue, CHE 205, Tampa, FL 33620, USA; eturos@usf.edu

**Keywords:** pathogenic bacteria, *Staphylococcus aureus*, MRSA, *N*-organothio β-lactams, *N*-thiolated β-lactams, antibacterials, thiolation, coenzyme A, covalent inhibitor prodrugs

## Abstract

Pathogenic bacteria such as the drug-resistant strains of *Staphylococcus aureus* dominate our medical and environmental landscape, causing hundreds of thousands of deaths from infections and life-threatening complications following surgeries. The availability of antibiotics and treatment protocols to control these microbes are becoming increasingly more limited as antibiotic resistance becomes more prevalent. In this article, a new family of small molecules referred to as *N*-organothio β-lactams is presented that have unique features and a mode of action against these pathogenic microbes, including multi-drug resistant strains, that may offer new options to address these concerns. This review gives an overview of the initial discovery, exploration and ongoing development of these synthetic antibacterial agents, with a focus on their unique properties and capabilities that provide fresh opportunities for combating pathogenic bacteria.

## 1. Introduction

Pathogenic microorganisms, including bacteria, fungi and protozoa, have been the targets of infectious disease researchers for many decades as a means to better control our public health and reduce morbidity, mortality and suffering [1]. The most notable bacterial pathogens of course include *Staphylococcus aureus*, an opportunistic microbe found commonly within the nasal passages and surgical settings, known for causing infections in soft tissue, on implanted biomaterials and within bone. Of greatest concern is that *S. aureus* is among the many genera of pathogens that are known to cause deadly infections that are increasingly multi-drug resistant and thus very difficult to treat or completely obliterate. A variety of methods have been devised to minimize the risk of infection by pathogenic microbes, particularly in medical centers where these microbes thrive, and interventive treatments with antibacterial agents. These latter approaches include small molecule antibiotics and anti-infectives, as well as the goal of identifying novel cellular targets and metabolic processes not yet investigated, to avoid the onset of acquired antibiotic resistance that makes the problem even worse. These aspects of drug discovery drive the underlying mission of our research on *N*-organothio β-lactams.

## 2. Summary of Research on *N*-Organothio β-Lactams

It is widely viewed that the β-lactam antibiotics, which include penicillins and cephalosporins, are the most significant class of antibacterial agents discovered [2]. The nitrogen center in these bicyclic β-lactams is sufficiently pyramidalized to disrupt planarity of the amide linkage, enhancing reactivity toward nucleophilic ring opening that makes them effective inhibitors of cell wall crosslinking proteins. This crosslinking involving the conversion of the D-alanine-D-alanine termini to a new interstrand amide is the final step of murein biosynthesis in bacteria, blocked by β-lactams via ring-opening acylation of the catalytic serine of the transpeptidase enzyme.



*N*-Organothio β-lactams are β-lactam compounds with an organosulfur substituent on the nitrogen and are generally monocyclic [3,4].



The first *N*-organothio-substituted β-lactams were described by Marvin Miller’s laboratory at University of Notre Dame with thiamazins, the sulfur analogs of oxamazins [5]. The Miller laboratory found that, unlike the bioactive oxamazins, thiamazins have no antibacterial activity, suggesting that the larger atomic radius of sulfur versus oxygen prevents effective binding within the bacterial transpeptidase active site.



Miller’s laboratory also reported the preparation of *N*-methylthio derivatives for utility as *N*-protected azetidinones in synthesis but, lacking the critical carboxylic acid functionality, did not investigate these as possible antibacterials. These compounds were stable to acid and base but readily cleavable by thiophiles to produce the *N*-protio lactam without opening of the four-membered ring.



In the early 1990s, our laboratory became interested in *N*-alkylthio β-lactams as synthetic precursors for studies on sulfur ring formation by electrophile-promoted cyclization of unsaturated thioethers [6,7]. In particular, we were examining various factors that might affect *exo* versus *endo* preferences, as well as relative stereochemistry of the ring closure.



One of the factors we hoped to examine was the presence of a conformationally strained ring, such as a β-lactam, at different locations on the chain, in order to ascertain if ring constraints could alter *exo* versus *endo* regiochemical pathways for the cyclization. By attaching a sulfur moiety onto the nitrogen center of a β-lactam, the cyclization of the sulfur center onto a neighboring electrophilically activated unsaturation could lead to the formation of *N*-fused penam or cepham ring structures.



As a prototype, we synthesized *N*-methylthio lactam **1** by a formal [2+2] coupling of the *N*-*p*-methoxyphenyl imine of phenylpropynal with methoxyacetyl chloride, providing exclusively the *cis* stereoisomer of the *N*-aryl-protected β-lactam. The *p*-methoxyphenyl group was then removed with aqueous ceric ammonium nitrate. The *N*-protio β-lactam was then treated with n-BuLi at −78 °C and methyl methanethiosulfonate to obtain the desired *N*-methylthio β-lactam **1**.



We then subjected β-lactam **1** to iodination and found that the reaction resulted in exclusively the desired 4.5-fused isopenem ring assembly [8]. By subsequently substituting the vinyl iodide, we were able to derivatize the rings with a carboxylic acid as a means to potentially enable binding to the penicillin-binding protein needed for transpeptidase activity. However, each of these isopenem derivatives we synthesized and bioassayed against common bacteria were completely inactive as growth inhibitors.



Through this cyclization strategy, we were able to access three alternative types of fused bicyclic β-lactam ring systems I–III [9,10].



### 2.1. Computational Studies

The relative stabilities of these three β-lactam-fused structures were examined computationally, with the N-S-fused ring system (type III) being more similar to the natural *N*-fused bicyclic rings of the penams. Ab initio calculations suggested that the N-S-fused ring is also more twisted than the classical penam ring, lowering its LUMO energy level while further enhancing the electrophilic character of the lactam carbonyl through electron withdrawal. These features led us to speculate that perhaps the isopenams and isopenems could possess antibacterial activity or utility as β-lactamase inhibitors [11].

### 2.2. Initial Biotesting of the N-Thiolated β-Lactams Against Common Bacteria

We looked for bioactivity by subjecting these monocyclic and bicyclic β-lactams to standard in vitro Kirby–Bauer antibacterial assays on agar plates against a variety of common Gram-positive and Gram-negative bacteria. The amount of material used for the testing ranged from 10 to 100 ug dissolved in DMSO solution (1 mg/mL). To our disappointment, none of the bicyclic β-lactam derivatives had any detectible in vitro bioactivity. Likewise, none of the monocyclic *N*-aryl or *N*-protio β-lactams that served as synthetic precursors were bioactive either. However, to our astonishment, monocyclic *N*-methylthio β-lactam **1** showed strong growth inhibition against *Staphylococcus aureus*, including various strains of methicillin-resistant *Staphylococcus aureus* (MRSA) [12].

### 2.3. Stability Studies of N-Methylthio β-Lactams

Solution studies indicated that the β-lactam ring of N-organothio β-lactams is stable in aqueous media and does not undergo hydrolysis or loss of the sulfur substituent in aqueous media from pH 3 to 10 (the lactams were pre-dissolved in DMSO). The stability of the azetidinone ring and the N-S bond as well was surprising given the noted instability of bicyclic β-lactams that is directly attributed to their antibacterial activity. Similarly, the compounds also had no activity in inhibiting the hydrolysis of penicillin G by commercial penicillinase and, in fact, were recovered unchanged from the solution. *The β-lactam ring in these compounds appears impervious to enzymatic cleavage by β-lactamases.* The inherent hydrolytic stability of these *N*-methylthio lactams both to acid and basic conditions, and to the enzymatic ring opening, was a bit surprising to us, and we therefore set out to further explore their antibacterial properties and capabilities in comparison to those of traditional β-lactam antibiotics like penicillin.



### 2.4. Exploring the Structure–Activity Features of N-Methylthio β-Lactams

Our initial structure–activity studies focused on examining the derivatives we already prepared for our synthetic work. Starting from lead structure **1**, we compared the bioactivity of the alkenyl β-lactam compounds **2** and **3** and acetylenic derivative **4**. All four of these *N*-methylthio lactams were similarly active against *S. aureus*.



From there, we examined *N*-methylthio β-lactams **5** and **6** with phenylethyl and phenyl substituents, respectively, at the C_3_ ring center. Both of these compounds possessed strong anti-*staphylococcal* activity as well.



These results clarified that all six of the *N*-methylthio β-lactams **1–6** were antimicrobially active and that the presence of unsaturation (e.g., alkene or alkyne) at C_3_ of the β-lactam ring was not a requirement for antibacterial activity. The significance of this finding was that we could rule out the requirement of forming a sulfonium ion intermediate such as structure II via the activation of the unsaturated side chain (intermediate I), which could further implicate these compounds as acting as in vitro alkylating agents.



Since this C_3_ side chain on the β-lactams seems to be variable, we elected to replace it for a wide variety of substituted aryl groups, including those with electron-donating or electron-withdrawing groups of many different types, located at alternative centers of the aryl ring [13].



Halogenated aryls had the greatest activity, consistently, regardless of which halogen, how many halogens, or even their locations on the aryl ring. Among all of these, however, we determined that the *ortho*-chlorophenyl moiety (β-lactam **7**) provided the most potent antibacterial activities [14]. Larger aryls, including naphthyl and fluorenyl, were also effective, albeit the increased size and molecular weight decreased the diameters of the growth inhibition zones on the agar plates, perhaps due to diminished solubility or agar mobility properties.

With the *o*-chlorophenyl β-lactam **7** being an optimal anti-MRSA compound, we next explored whether other, larger *N*-alkylthio groups could be acceptable. We examined this by changing the *N*-methylthio group for different organothio residues by systematically increasing the alkyl chain length, as shown for β-lactam analogs **7–11**.



For these five derivatives, the anti-MRSA activity increased for the *N*-ethylthio lactam **8**, but further elongating the chain, the activity incrementally dropped [15]. Thus, *N-n*-octylthio β-lactam **11** had less than half the anti-MRSA activity of *N*-methylthio lactam **7**. To ensure that the diminishing antibacterial activity was not simply due to chain elongation causing a steady decrease in the β-lactams ability to diffuse through the agar gel, thereby resulting in smaller zone sizes, we repeated the assays by measuring the minimum inhibitory concentrations in broth. We found that the Kirby–Bauer growth inhibition zone sizes mirrored closely with the MIC values, and we confirmed that either assay method was reliable for determining the initial trends in bioactivity within a family of closely related structures.

With ethyl and propyl being the optimal chain lengths for the alkylthio group, we next explored the effect of branching by preparing and assaying *N*-organothio β-lactams **12–17** for anti-MRSA activity.



Among these six variants, the *N*-isopropylthio and *N*-*sec*-butylthio analogs **12** and **13** afforded the best anti-MRSA activity, even surpassing that of the *N*-methylthio or *N*-ethylthio β-lactams (**7** and **8**). Interestingly, *N*-cyclohexylthio, *N*-phenylthio and *N*-benzylthio β-lactams **15–17** were all much less active. Of all these alkylthio residues, we determined that the *N-sec*-butylthio had the strongest bioactivity, at least towards MRSA. With establishing the need for a sulfur group on the lactam nitrogen, we moved on to investigate the role of substituents at the two carbon centers of the ring [16].

We began with compounds lacking the C_2_ methoxy substituent entirely and were again surprised to observe that the C_3_-*unsubstituted* β-lactams had only very weak or no anti-MRSA activity.



We also prepared several C_2_-chloro, iodo and azido β-lactam derivatives. Among these, the C_2_-chloro lactam had the strongest activity against several MRSA strains, with the zones of growth inhibition being more than double that of penicillin G and equivalent to C_2_-methoxy lactam **7**.



We next prepared some C_2_-amino-substituted analogs, each obtained as the *trans* isomer.



These C_2_-amino-substituted β-lactams were considerably less potent against MRSA than the C_3_-methoxy β-lactam compound **7**, and of these, only the *N*-benzylamino derivative afforded appreciable bioactivity. We considered that the increased side chain polarity might account for this drop in activity. However, even more unexpectedly, substituting the methoxy for a phthalimidyl, benzylthio or simple ethyl group resulted in compounds with little to no bioactivity towards MRSA.



C_2_-Alkoxy, C_2_-phenoxy, C_2_-acetoxy and C_2_-hydroxyl β-lactams were synthesized and their bioactivities compared. While all of these analogs showed stronger anti-MRSA activity compared to penicillin, none were as potent as C_2_-methoxy β-lactam **7**.



The three *O*-sulfonated lactam analogs also showed anti-MRSA activity that, curiously, increased with the increasing size and lipophilicity of the sulfonate moiety (methyl<phenyl<p-tolyl).



Noting that all of the previous derivatives were monosubstituted next to the lactam carbonyl, we next investigated the effect of disubstitution at C_2_ with the compounds shown below. The propyl and allyl ethers were more active than the more polar methoxymethyl derivative. Antibacterial activity decreased as steric congestion was increased. For the four acetoxy esters, the methyl and propyl analogs were more active than the allyl- or phenyl-substituted derivatives.



Additionally, a short series of C_2_-spirocyclic compounds were prepared and tested. All of these spirocyclic derivatives had reasonable anti-MRSA activity but were at least 25% less active than the corresponding C_2_-open ring analogs above.



Finally, we noted that the *cis* isomer of C_2_-acetoxy *N*-methylthio β-lactam was slightly less active against MRSA than the *trans* stereoisomer.



Finally, the dependency of the β-lactam’s *absolute* stereochemistry on anti-*staphylococcal* activity was evaluated by examining both enantiomers, individually prepared by enzymatic kinetic resolution of the *N*-aryl acetoxy lactams. Each enantiomer had precisely the same activity against MRSA, producing identical zones of growth inhibition on agar plates and minimum inhibitory concentration (MIC) values in broth microdilution assays.



While these structure–activity profiles indicate clearly that the relative and absolute stereochemistry of the azetidinone ring do not influence the antibacterial activities of the *N*-methylthio β-lactams, there are some subtle differences in this, such as 2,3-*trans* analogs being marginally more active than the *cis* isomers. Finding that the *N*-*sec*-butylthio moiety provides higher anti-MRSA bioactivity compared to the *N*-methylthio- or *N*-ethylthio-substituted analogs, the four *N*-*sec*-butylthio β-lactam diastereomers were synthesized and examined individually.



Bioassays showed that the two 2*S*,3*R* β-lactams and the two 2*R*,3*S* compounds had equivalent anti-MRSA activities, but the 2*R*,3*S* diastereomers were 10–15% more bioactive. Granted, these differences in bioactivity are not all that significant but indicate perhaps that the chirality of the two lactam centers affects the bioactivity when the *N*-organothio group itself also contains chirality.

### 2.5. Role of the Organothio Substituent on anti-MRSA Bioactivity

While all of the β-lactams above with antibacterial activity are *N*-sulfenylated compounds, we wondered whether the oxidation state of the sulfur may also play a role [15]. For this, *N*-sulfinyl and *N*-sulfonyl β-lactam analogs were synthesized by treating the *N*-cyclohexylthio precursor with 30% hydrogen peroxide in glacial acetic acid. The *N*-sulfinyl β-lactam was obtained as a 1:1 diastereomeric mixture, while the *N*-sulfonyl β-lactam was prepared by further oxidization with excess hydrogen peroxide. Within this series of three compounds, the *N*-sulfenyl and *N*-sulfinyl β-lactams each showed anti-MRSA bioactivity, but the *N*-sulfonyl analog was inactive.



We further found that the most highly oxidized derivatives, *N*-sulfonate salt and its *N*-sulfonic acid (prepared independently, as shown below), were both completely inactive against MRSA.



These non-active *N*-sulfonic acid lactams are structurally similar to the biologically active *monobactams* (monocyclic beta-lactams), such as aztreonam, which have broad activity against aerobic Gram-negative bacteria but little or no activity against Gram-positives such as *Staphylococcus aureus* [17].



Overall, this collection of structure–activity data reveals that the β-lactam unit is not a requisite for MRSA (or bacterial) bioactivity, but the *N*-sulfur moiety is, and that the *S*-oxidation state, chirality and type of substituents at the two lactam carbon centers have a more subtle role on the observed bioactivity. This is summarized below.



Although we did not define the biological or chemical mode of action or the putative cellular target, the SAR suggests a need to balance the lipophilic character of the substituents to obtain an optimal anti-MRSA activity, with C_2_-alkoxy or acyloxy side chains and C_3_ aryl moieties with the best antibacterial activity. Both highly polar and nonpolar substituents diminish antibacterial activity. This speaks to the requirement for the *N*-thiolated β-lactam to pass through the bacterial cell membrane more than to a specific non-bonding interaction with a cellular target. This alone is unusual, since all other known β-lactam antibacterials exert their effects from the outside of the cell, where the penicillin-binding proteins are present, and do not require entering the cell at all.

### 2.6. Bacterial Selectivities

The bulk of our early studies of *N*-thiolated β-lactams focused on their structure–activity towards *Staphylococcus* bacteria—in particular, MRSA. The data indicated some critical new insight about not only the structural requirements but also possible chemical mechanisms for their bioactivity. Most significantly, the *N*-organothio moiety is an absolute requirement for bioactivity, while the other ring substituents alter efficacies. To extend these investigations further, we set out to evaluate what other microbes besides *S. aureus* might be susceptible. We surveyed a more extensive collection of Gram-positive and Gram-negative bacteria. For most of these 23 genera, more than one species or strain was evaluated, showing a more global perspective as to the breadth of activity of the lactams and which genera are most sensitive as determined by Kirby–Bauer disk diffusion assays on agar plates. Bacteria with the highest susceptibility to the β-lactams produced large growth inhibition zones, while others with no zones were unresponsive to the lactams. Although in vitro activity varied considerably among the individual β-lactam derivatives, the overall trends were very consistent with respect to which microbes were susceptible. The strongest activity was uniformly observed for *Staphylococcus aureus* (including both non-clinical and clinical MRSAs), *Micrococcus* and *Bacillus anthracis.* Beyond these, a few microbes were moderately sensitive, such as *Neisseria gonorrhoeae, Bacteroides* and *Streptococcus,* while *Salmonella typhimurium*, *Vibrio cholerae* and *Mycobacterium tuberculosis* were appreciably less responsive. None of our β-lactams were active against *Enterobacter cloacae*, *Escherichia coli*, *Klebsiella pneumoniae*, *Listeria monocytogenes*, *Bacteroides fragilis*, *Serratia marcescens*, *Pseudomonas aeruginosa* or *Proteus mirabilis*.

This illustrates just how selective these β-lactams are towards, in effect, just nine genera in total from the dozens of bacteria tested, and most curiously for the compounds in the β-lactam families, this activity did not appear to be related to whether the microbe is Gram-positive or Gram-negative. Those most affected by the *N*-thiolated β-lactams are *Bacillus*, *Staphylococcus* and *Streptococcus*. However, *Enterococcus*, *Lactococcus* and *Listeria* were found to be unresponsive to the β-lactams. Within the staphylococci grouping, we found broad bioactivity for most of the β-lactams against *S. aureus* (MSSA and MRSA), *S. epidermidis* and *S. lugdunensis* but not against *S. lentus* and *S. simulans*. Therefore, the lactams do not affect all *staphylococci*. We examined seven species of *Bacillus*, including *B. anthracis*, *B. globigii*, *B. thuringensis*, *B. megaterium*, *B. coagulans*, *B. subtilis* and *B. cereus* [18]. Inspection of the structure–activity profiles indicated that lipophilic acyloxy or alkoxy groups at C_2_ of the β-lactam ring afford the best growth inhibition properties against the seven *Bacillus* species examined, with pro-poxy and allyloxy side chains being a bit more active than the C_2_-methoxy lactam. Spirocyclic ethers at C_2_ were less active than the open chain variants. suggesting that *Bacillus* may be more sensitive than *Staphylococcus* to lipophilicity within the C_2_ alkoxy side chain. The *N*-organothio moiety was again required for anti-*Bacillus* activity, with the *sec*-butylthio compound **13** having the best overall bioactivity against MRSA. In fact, this analog had zone sizes about double in diameter to those of *N*-methylthio β-lactam **7**. The broth MIC value for lactam **7** against both the avirulent Sterne and virulent Ames strains of *B. anthracis* was 0.5 µg/mL. The structure–activity trends were remarkably similar to those observed against the staphylococci, suggesting that the mode of action of these lactams is likely closely related in *Bacillus* and in *Staphylococcus.* These unusual bacterial selectivities and the structure–activity profiles suggested a unique mode of action. We then wanted to ascertain whether these β-lactams act as bacteriostatic or bactericidal agents. Cell survival experiments using methicillin-susceptible *S. aureus* (MSSA) and MRSA, respectively, were conducted in the absence versus the presence of β-lactam **13** over a 2-h growth window. In the absence of β-lactam **13**, MSSA and MRSA each grew logarithmically, while, in the presence of β-lactam **13**, bacterial growth was immediately stopped, but the number of viable cells at or above the MIC of the β-lactam did not diminish. This indicated a bacteriostatic effect, not cidal, as is the case for the classical β-lactams.

The observation that *N*-thiolated β-lactams are bacteriostatic and act on only a narrow subrange of bacteria suggested that their mode of action is distinct to all other β-lactams and that they may act upon biological targets inside the cytoplasm of the bacterial cell. Various antibacterial compounds that target processes intracellularly, such as protein synthesis (chloramphenicol) or the production of secondary metabolites (sulfa drugs), are primarily bacteriostatic, while those acting extracellularly on or just outside the bacterial membrane (polymyxins, bacitracin and cationic lipoproteins); cell wall (penicillin and vancomycin) or DNA (metronidazole) are bactericidal. By identifying whether these lactams affect a primary cellular process (protein, nucleic acid or fatty acid biosynthesis), we hoped to gain a better understanding of their mechanism of action [19].

### 2.7. Stability of the N-Thiolated β-Lactams to β-Lactamases

One of the curious findings from the bioassays was that the β-lactams are uniformly active against all the *S. aureus* strains, including the MRSAs. This is in stark contrast to penicillin, which bioactivity drops drastically for the penicillinase-producing strains of MRSA. We therefore deemed it worthy to examine the influence of penicillinases on the activity of our lactams. The unusual SAR trends show a preference for lipophilic side chain residues, not ionic functionality on the framework of the *N*-thiolated β-lactam, that binding to and subsequent ring opening by β-lactamases are unlikely. We confirmed the stability of the *N*-thiolated β-lactams to penicillinases, both in buffered solutions as a means to possibly inhibit the catalytic function of the penicillinase protein and in competition experiments in which the lactams are co-mixed with penicillinase during Kirby–Bauer in vitro experiments. These studies convincingly showed that *N*-thiolated β-lactams do not act as β-lactamase inhibitors and have no ability to impede the enzymatic hydrolysis of penicillin G by penicillinases. The growth inhibition zones and the MIC values for the lactams do not change when penicillinase is added to the culture broth or agar plates, indicating that the β-lactams are impervious to β-lactamase ring openings. *This feature makes these molecules unique among β-lactam antibacterials and potentially valuable.*

### 2.8. Examining if the N-Thiolated β-Lactams Could Alkylate DNA in Bacteria

We were quickly able to discount the role of the lactams as possible alkylating agents towards cellular nucleophiles such as DNA. We considered this possibility when recognizing that the β-lactam *N*-organothio substituent could conceivably transfer the organo chain to a biological nucleophile via alkylation, with loss of the *N*-thiol β-lactam. However, this species was never isolated in our culture experiments. Furthermore, the initial studies show that the *N*-ethylthio, *N*-isopropylthio and *N-sec*-butylthio lactams are increasingly *more active* against *S. aureus* than the *N*-methylthio derivative, whereas an alkylation mechanism would predict the exact opposite trend. Furthermore, should the β-lactams alkylate DNA, this would be expected to inhibit DNA replication or compromise its structural integrity and lead to apoptosis. Examples of bacterial DNA alkylating agents (leinamycin and mitomycin) are cidal in their activity, while the *N*-thiolated lactams are bacteriostatic, so this would seem to rule this possibility out. We also conducted experiments to determine if *N*-thiolated β-lactam **1** could induce duplex strand breakage of supercoiled DNA by treating the plasmid pBR322 at varying concentrations of β-lactam **1** in sodium phosphate buffer for 24 h. Double-strand (fragmentation) and single-strand (linearization) breakages were then analyzed by agarose gel electrophoresis containing 1% ethidium bromide. We ascertained that, at these high concentrations (5–100 μM), the β-lactam failed to induce fragmentation or relaxation of the plasmid superhelix, further indicating that there is no direct chemical modification of the DNA by the lactams. Given that some DNA alkylators require chemical activation, often by a thiol-mediated reaction acting on a disulfide or trisulfide precursor, we also illustrated that *N*-methylthio β-lactam **1** in the presence of added glutathione, dithiothreitol or β-mercaptoethanol still exerted no effect on the plasmid. From this, we can conclude that these β-lactams do not appear to alkylate bacterial nucleophiles to the extent expected to account for their antibacterial properties.

### 2.9. Examining Effects of the N-Thiolated β-Lactams on Bacterial Cells by Scanning Electron Microscopy

It is widely known that antibacterial compounds such as the penicillins that block cell wall biosynthesis in bacteria, or act on the cytoplasmic membrane (i.e., polymyxins), cause serious deformations in cell morphology and on the outside surface of the cell wall that can be observed readily by scanning electron microscopy (SEM). We tried to visualize possible alterations in MRSA cells cultured in the presence of an *N*-thiolated β-lactam using SEM but excised regions in the agar from the Kirby–Bauer well diffusion plates along the outermost periphery of the growth inhibition zones, where antibiotic concentrations were insufficient to cause an inhibitory effect. However, these regions were expected to contain cells that survived exposure to the lactam. SEM showed that the remaining cells were undamaged and grew in their normal grape-like clusters, with uniform spherically shaped cell structures. Conversely, we observed *S. aureus* surviving exposure to penicillin G, along these same sub-lethal regions in the agar, was severely morphologically deformed, indicative of the action of penicillin on cell wall biosynthesis. The untreated and β-lactam-treated *S. aureus* cells each took on the purple coloration upon Gram-staining, characteristic of intact, undamaged cell walls, while penicillin-treated *S. aureus* cells stained light pink, reflecting the damage inflicted by the drug on the murein. These SEM experiments confirm that the *N*-thiolated β-lactams exert their static effects on growth without altering cell morphology or the integrity of the cell wall, the manner in which penicillin or other β-lactams interfere with murein assembly.

### 2.10. Effect of the N-Thiolated β-Lactams on Radioisotope Uptake into S. aureus Cells

Our early investigations into trying to understand the cellular effects of these lactams focused on radio uptake experiments. This recognizes that traditional bioactive molecules may alter transcription or translation into proteins. Our experiments tracked the uptake of radiolabeled ^3^H-uridine into RNA when culturing *S. aureus* with varying concentrations of the β-lactam versus rifampicin, a potent RNA synthesis inhibitor. We observed that the β-lactam slightly altered ^3^H-uridine incorporation into bacterial cells but not to the extent of rifampicin, ruling out nucleic acid transcription as a primary target. To further corroborate this, pulse labeling experiments were carried out to track ^3^H-thymidine uptake during DNA replication in the presence versus the absence of β-lactam. The control antibiotic ciprofloxacin inhibited the radiolabeled thymidine uptake, but lactam **3** (or penicillin G) did not. Alternatively, we turned to protein synthesis as a potential target, and again, the β-lactam had minimal effect on the uptake of ^3^H-isoleucine in *S. aureus* when treated with either β-lactam or chloramphenicol (a translation inhibitor). The data revealed that the lactams exert a delayed response on protein synthesis but, again, was not a primary basis for the observed antimicrobial activity of the β-lactams.

Next, we examined fatty acid biosynthesis in *S. aureus*, and here, we observed an immediate effect on the rate of uptake of radiolabeled ^3^H-acetate in cells treated with β-lactam. The interruption of radioisotope incorporation into the cell in the presence of *N*-methylthio β-lactam, but not penicillin or DMSO controls, clearly indicated that the *inhibition of fatty acid biosynthesis is a primary antibacterial effect* in *S. aureus*.

### 2.11. Metabolic Fate of the N-Thiolated β-Lactams in S. aureus

To investigate possible cellular targets of these lactams in bacteria, we exposed a broth culture of *S. aureus* with an *N*-methylthio β-lactam. After several hours, the dethiolated *N*-protio β-lactam was extracted from the media with ethyl acetate.



This shows the facility in which β-lactam transfers the *N*-organothio entity to a bacterial thiophilic species, the most common being glutathione, coenzyme A, bacillithiol [20] and mycothiol, that make up the native redox buffer in various cells, responsible for maintaining homeostasis of the oxidation states of proteins while removing cytotoxic entities such as alkylating agents and reactive oxygen species from the cytoplasm.

### 2.12. Attenuation of Bioactivity of N-Thiolated Lactams by Cytosolic Thiols

We were able to demonstrate the sensitivity of the *N*-methylthio β-lactams to glutathione by co-mixing glutathione with β-lactam during the Kirby–Bauer well diffusion experiments. The amounts of glutathione added to the lactam in this mixture were varied to prove that the bioactivity was dependent on the degree by which the glutathione could dethiolate the lactam and thus reduce its antibacterial capabilities. The greater the amount of glutathione, the lower the MRSA bioactivity of the *N*-thiolated lactam. Further, the thiol susceptibility was illustrated to depend on the structure of the alkylthio group. For this experiment, four 6-mm diameter wells were cut into an agar plate, one being in the very center and three equidistant around it. The center well was charged with 1 mg of glutathione as an aqueous solution, and the three β-lactams were then added in DMSO to individual wells on the perimeter, equidistant from the center well containing glutathione. After 24 h, we observed that the growth inhibition zones appearing around the wells of the three β-lactams were all noticeably “indented” on the side facing the central well. This indicated that the outward diffusion of glutathione from the center well reduced the growth inhibitory effects of the β-lactams on the microbe in all three cases. We ran a control plate wherein the center well contained other compounds, such as a DMSO blank, or different amino acids. Thos plates showed no perturbations in the shapes or sizes of the growth inhibition zones.

To further corroborate these findings, we carried out a different set of experiments, in which the same three *N*-alkylthio lactams were exposed to added glutathione in the growth media under bacterial growth conditions—in this case, varying the amounts of glutathione added to the growth media. We observed that *N*-ethylthio lactam and *N*-*sec*-butylthio lactam gave their full antibacterial activity against *S. aureus* in the presence of a small amount (20 mg) of glutathione added into the growth media, while *N*-methylthio lactam **7** was completely inactivated. This indicated that the *N*-methylthio derivative was most rapidly susceptible to the negating effect of glutathione. Increasing the amount of glutathione in the agar showed that now both the *N*-methylthio and the *N*-ethylthio β-lactams completely lost their bioactivity, while the *N-sec*-butylthio compound was still strongly active. Thus, the three *N*-alkythio lactams do not have the same stabilities to glutathione (or possibly other cellular thiophiles), with the order of stability being *N*-methylthio < *N*-ethylthio < *N-sec*-butylthio. This difference in glutathione susceptibilities in the presence of *S. aureus* mirrors the in vitro activities towards *S. aureus* (including MRSA).

From these investigations, we concluded that elevated concentrations of glutathione in the cytoplasm of many microbial, as well as human, cells effectively neutralize the biological effects of these *N*-thiolated β-lactams. This likely rationalizes the differences in selectivity observed for *S. aureus* and certain other microbes (such as *Bacillus*) in our in vitro assays. *E. coli* and *P. aeruginosa* are two of the bacteria with high concentrations of cytosolic glutathione that are not susceptible to the β-lactams.

### 2.13. Studies on the Effect of N-Thiolated β-Lactams on the Bacterial Redox Buffer

Living cells including bacteria maintain their requisite redox state by establishing a robust redox buffer system [20]. This is required to maintain stasis in the cell, as well as to intercept undesired chemicals that are detrimental to cellular components, such as alkylating agents and reactive oxygen species that are highly damaging to proteins, nucleic acids and cofactors. The redox buffer consists of an equilibrium mixture of a small thiol molecule (as the reducing agent) and its disulfide counterpart (as the oxidant), and the concentration of free thiol in these cytosolic buffers is typically high (often millimolar). These thiols are glutathione, coenzyme A or mycothiol, depending on the bacterial species. In *Staphylococcus*, it is coenzyme A that is dominant, with little to no glutathione, and the relatively high level of reduced CoA (as the free thiol) versus oxidized CoA (CoA disulfide) is maintained by CoA disulfide reductase, a protein that itself contains a cysteine residue, as well as a non-covalently bound flavin moiety. Reduction of CoA disulfide to the reduced CoA thiol occurs through thiol-disulfide exchange with the active site cysteine, followed by flavin-mediated hydride transfer from NADPH to reduce the cysteine-CoA disulfide bond and regenerate the active site. In other cells, including mammalian, the glutathione-glutathione disulfide redox equilibrium is controlled by glutathione disulfide reductase.

### 2.14. Studies on the Effect of N-Thiolated β-Lactams on the Bacterial Disulfide Reductase Enzymes

Despite some similarity, coenzyme A disulfide reductase differs in its mechanism and substrate (disulfide) selectivity to glutathione disulfide reductases that regulate glutathione-based redox. The correlation between the antibacterial activity of the *N*-alkylthio β-lactams and the CoA-based redox in *S. aureus* (and other microbes) led us to investigate whether the lactams were interacting with the catalytic capabilities of CoA disulfide reductase. In particular, we suspected that the catalytically active cysteine thiol of the reductase could be sulfenylated by alkylthio transfer from *N*-thiolated β-lactam. However, our experiments determined that neither lactam nor CoA-alkyl disulfide formed from lactam reacting first with cytosolic CoA had any measurable inhibitory effect on CoA reductase activity. Thus, it appears that, even if CoA reductase’s cysteinyl group is thiolated, the formation of this mixed alkylthio-protein disulfide does not permanently alter the enzyme function, as is likely rapidly reduced back to the catalytically active thiol form. This rules out the proposition that the lactams act on the CoA disulfide reductase enzyme as a primary basis for its bioactivity. Given this finding, and the realization that the CoA thiol-redox buffer of *S. aureus* remains unaltered by the *N*-alkylthio β-lactams, another possibility we considered was that the lactams may transfer the alkylthio group directly to the thiol group of coenzyme A, giving a mixed CoA-alkyl disulfide. If so, this disulfide may be expected to, in turn, bind and transfer the alkylthio moiety to sulfhydryl groups of CoA-dependent enzymes involved in bacterial fatty acid biosynthesis. The most likely one is an enzyme called β-ketoacyl-acyl carrier protein synthase III (FabH). The Reynolds laboratory have studied FabH extensively and have reported that the formation of mixed CoA-alkyl disulfides (prepared synthetically) in solution inhibits the enzymatic activity of FabH by sulfhydryl thiolation but exerts no antibacterial activity when tested in cyto [21]. They suggested that these highly polar CoA-alkyl disulfides are unable to pass through the bacterial membrane into the cytosol. This led us to propose that our lipophilic *N*-alkylthio lactams can penetrate easily through the cell membrane into the cytoplasm, where they can transfer the *N*-alkythio residue onto CoA in situ, and this mixed CoA-alkyl disulfide species can now act directly on FabH intracellularly. *This is our current hypothesis on the mode of action of N-thiolated β-lactams.* Moreover, the selectivity these disulfides display towards a particular FabH protein is highly dependent on the structure of the alkyl residue of *N*-alkylthio β-lactam, which might provide a reason for the variation in antibacterial activities seen among different *N*-alkylthio β-lactams and different microbes.

### 2.15. Identifying a Cellular Target of the N-Thiolated β-Lactams in S. aureus

Tracking the glutathione sensitivities in cells (and in growth media) versus the measured microbiological activities for each of our synthetic *N*-thiolated β-lactams showed us an appreciation for the unique features of these compounds while at the same time acknowledging that we still did not know the underlying basis for the antibacterial properties in those bacteria happening to express low (or no) glutathione. This motivated us to try to identify the primary target in bacteria responsible for biological properties, including the experimentally observed effects on lipid (primarily) but not protein or nucleic acid biosynthesis from the radio uptake experiments. We decided to approach this using a polymer-bound β-lactam to search for potential targets in bacterial fluid extracts. This resin-bound lactam was synthesized with Merrifield resin and then added directly into a broth of *S. aureus* lysate. After a period of time, the lysate was extracted with ethyl acetate from which was isolated the expected *N*-protio lactam. This confirmed the release of the lactam from the polymer by nucleophilic scission of the N-S linkage. The solid component from the media, presumed to be a mixed disulfide of the alkylthio moiety attached to the cellular target molecule, was treated with diisobutylaluminum hydride, and the organic media was filtered, then evaporated. HPLC analysis of the evaporate produced a single compound identified by mass spectrometry and proton NMR as coenzyme A. Thus, we conclude that *coenzyme A is a primary cellular target of N-thiolated β-lactams* [19].



Coenzyme A is a low molecular weight thiol in the cytoplasmic fluid of *S. aureus* and several other important pathogenic microbes, including *Bacillus* and *Micrococcus* species. The bacteria expressing the highest concentrations of cytosolic coenzyme A—namely, *Staphylococcus*, *Bacillus* and *Micrococcus*—also have the lowest concentration of glutathione in the cytoplasm and show the highest susceptibility to the *N*-thiolated β-lactams.



We still cannot unequivocally say that the lactams react directly with either coenzyme A or glutathione in the cytoplasm, as we have observed that thiol additives, including coenzyme A, glutathione and cysteine, neither react with nor neutralize the biological effects of the *N*-thiolated lactams, at least in pH neutral buffer solution. The transfer of the organothio moiety seems to require the presence of cultured bacterial cells, which indicates the likelihood of a transferase protein or other cellular species in bacteria that may mediate the transfer of the alkylthio group onto the thiol of CoA or to other enzyme-bound thiols.

### 2.16. Effect of Exogenous Fatty Acids on the Antibacterial Activity of the N-Thiolated β-Lactams

The presence of fatty acids such as oleic acid and Tween 80 in bacterial growth media has been reported to overcome the inhibitory effects of compounds that prevent fatty acid biosynthesis. In this manner, bacteria which fatty acid production is impeded can replace missing fatty acids from the external environment. We therefore examined the addition of exogenous fatty acids to reverse the antibacterial effects of the lactams against MRSA USA 100, a particularly deadly strain of *S. aureus* in hospitals. We measured the MIC of *N-sec*-butylthio β-lactam 7 in the presence of oleic acid and Tween 80 versus in their absence and compared the results to those of bovine serum albumin, vancomycin (a cell wall biosynthesis inhibitor) and triclosan (a FAS inhibitor). The MIC of β-lactam 7 showed only a very small increase in the presence of either fatty acid additive, while the MIC of vancomycin was unaffected by the supplementation of growth media with fatty acids, and, for triclosan, the MIC value increased 100-fold and 500-fold in the presence of oleic acid and Tween 80, respectively. This suggests that the lactams may produce a collection of inhibitory effects on fatty acid biosynthesis not easily circumvented, as is found for other fatty acid biosynthesis inhibitors such as triclosan that act solely on one major pathway.

In polyketide synthesis, chain growth involves Claisen condensations between adjoining thioesters, leading to continual chain growth of the polyketide. The continual incorporation of acetyl residues from *S*-acetyl coenzyme A ensures the lipid chain grows two carbons at a time.



The results of our studies on the *N*-thiolated β-lactams enable us to speculate that the thiols of the acyl carrier protein normally acylated are instead thiolated by the CoA disulfide, thus terminating chain growth.



These studies suggest that *N*-thiolated β-lactams are not only completely unique among β-lactam antibacterial agents in their structure–activity profiles, stability towards hydrolytic cleavage by β-lactamases and cellular mode of action as bacterial FAS inhibitors against select species of pathogenic bacteria. Consequently, *N*-thiolated β-lactams may hold significant potential for the treatment of deadly bacterial infections caused by MRSA or other pathogenic microbes.

### 2.17. Investigating Multi-action N-Thiolated β-Lactams

During his dissertation studies, graduate student Biplob Bhattacharya prepared a dual-action derivative **19** comprised of an *N*-sec-butylthio β-lactam esterified with *N*-acetylciprofloxacin [22]. Testing of compound **19** against a multidrug-resistant USA 300 clinical strain of MRSA found a MIC value below 0.0125 ug/mL (the lowest concentration we evaluated). It is also noted that attempts to culture surviving cells treated with this compound that express enhanced resistance were unsuccessful. The goal was to isolate surviving clusters of cells and to grow them exponentially to explore the basis for their survival or their induced resistance and the cellular mechanism(s) used to overcome the dual-action effect of **19**. This result points to the *prospect of preparing a wide range of dual- or tri-action antibacterial prodrugs linking multiple types of antibiotics that have distinct cellular targets and modes of action and which may markedly lower the incidence or the rate of resistance developing*. We anticipate that, in the future, this could be an effective way to broaden activity across more species of pathogenic bacteria while potentially reducing antibiotic resistance.



### 2.18. Nanoparticle-Bound Systems

In an effort to improve water solubility without sacrificing bioactivity, we explored some options for introducing more polar side chains onto the periphery of the β-lactam ring. Unfortunately, all of our attempts to come up with water-solubilized derivatives diminished in vitro antibacterial activities. It seems apparent that increasing polarity leads to lower cell membrane permeability and, thus, decreased bioactivity. Therefore, we turned to a different option of preparing polyacrylate nanoparticles of the *N*-thiolated β-lactams by aqueous emulsion polymerization.

We successfully implemented this methodology by mixing the *O*-acryloyl β-lactam in a 7:3 composition of butyl acrylate:styrene and inducing a radical polymerization with potassium persulfate in the presence of a small amount of sodium dodecyl sulfate [23]. The emulsion produced nanoparticle polymers uniformly measuring 45 nm in diameter in aqueous solution, onto which was covalently incorporated the β-lactam. We could also implement this to prepare emulsified nanoparticles in which the β-lactam unit is sequestered non-covalently within the polymeric matrix and thus be fully water-dispersed.



Testing against MRSA grown in agar media confirmed that both types of nanoparticles retained their antibacterial properties, even if coenzyme A or glutathione was added to the growth media in an attempt to reduce the activity of the lactam. These experiments illustrated that the emulsified nanoparticles not only allow the water-insoluble lactams to be deliverable in water but also that the nanoparticles afforded protection from damage by thiophilic agents in the media. Additional studies on these polyacrylate nanoparticle systems utilized carbohydrate sectors and various surfactant-bound constructs [24,25].

### 2.19. Extension to Related Thiophilic Antibacterial Agents

Recognizing that the β-lactam component of the *N*-thiolated β-lactams is not a strict requirement for biological activity and may merely present an appropriately activated organothio electrophile to coenzyme A inside the bacterial cell, we considered other possibilities for producing antibacterial compounds that might be able to achieve the same outcome and effectiveness. We first examined the replacement of the β-lactam with an oxazolidinone ring, being sure to maintain the transferable organothio moiety on the carbamate nitrogen center [26]. Thus, we came up with a variety of ring-substituted *N*-thiolated oxazolidinones **20** having similar functionality and substituents as that of the *N*-thiolated β-lactams.



The thio-transfer processes and resulting antibacterial, antifungal and anticancer activities associated with the *N*-thiolated β-lactams and *N*-thiolated oxazolidinones are highly reminiscent of those reported for the natural disulfides allicin and ajoene obtained from fresh extracts of garlic and onions [27]. These thiolating species express fairly widespread bioactivity as antibacterials, antifungal and, anticancer compounds and are even purported to be antiviral and antiparasitic. Like the *N*-thiolated β-lactams, allicin shows the strongest activity against bacteria having low levels of glutathione and blocks the biosynthesis of fatty acids and is susceptible to the neutralizing effects of cellular thiols, including cysteine, glutathione and coenzyme A [28].



We next turned to acyclic compounds that could deliver an electrophilic alkylthio moiety, replacing the lactam nitrogen with a different heteroatomic substituent that could serve as the leaving group upon nucleophilic replacement. In 2008, we reported the first of these, alkyl aryl disulfides [29], and, in 2012, structurally novel *S,S′*-bis(heterosubstituted) disulfides [30] (X=OR, SR, NHR or NR_2_), each with potent antimicrobial activities mirroring those of the *N*-thiolated β-lactams and *N*-thiolated oxazolidinones. These extend the range of functionalities in small molecules that can have similar utility as antibacterial compounds for pathogenic microorganisms [31].



The Long laboratory at Marshall University has reported an extension of this methodology to pyridyl disulfides intended to mimic the reactivity of allicin from garlic [32]. They found antimicrobial activity against vancomycin-intermediate and vancomycin-resistant *Staphylococcus aureus*, which were the most susceptible to disulfide derivatives with alkyl chain lengths of seven to nine carbons. These disulfides synergize with vancomycin against vancomycin-resistant *S. aureus,* inducing dispersion of *S. aureus* biofilms and slowing metabolic processes in *S. aureus* cells. They suggested that these pyridyl disulfides may have potential as adjuvants in the combination therapy of *S. aureus* infections.



The Long group at Marshall University described *N*-thiolated fluoroquinolones as having potent bioactivity against MRSA [33]. In this case, as we have found with our lactams, the *N*-propylthio substituent provides excellent bioactivity. The bioactivity was reliant on increased cell permeability and reaction with cytosolic thiols that generate mixed disulfides in bacteria, leading to the generation of reactive oxygen species in the cytosol.



In addition, they investigated the basis of antibacterial action of the alcohol sobriety aid disulfiram using a combination of differential transcriptomic, metabolomic, bioenergetic and phenotypic growth analyses [34,35]. Thiophilic compounds in the cytosol cleave the disulfide bond in disulfiram to form mixed disulfide conjugates that can chelate metals to form complexes. Transcriptome analysis indicated that disulfiram increases oxidative stress and redox imbalance in bacterial cells, leading to metal acquisition and an increase in the biosynthesis of pantothenate, coenzyme A, thiamine, menaquinone, siderophores/metallophores and bacillithiol. Metabolomic analysis showed that disulfiram depletes coenzyme A and affects the catabolism of glucose, pyruvate and NADH while upregulating arginine catabolism citrate consumption attributed to siderophore biosynthesis. The bioenergetic studies further revealed that the primary metabolite of disulfiram is likely involved in the mechanism of action as an inhibitor of oxidative phosphorylation and a chelating agent of iron and other metals. All in all, disulfiram inhibits *S. aureus* growth by altering the glucose catabolism and redox balance associated with heightened oxidative stress.



### 2.20. Antifungal Activity of N-Thiolated β-Lactams

Owing to the mechanism through which classical β-lactam antibiotics such as penicillins and cephalosporins act on bacterial cell wall crosslinking by transpeptidases, eukaryotic cells such as fungi would not be expected to be affected. However, the unusual structure–activity, morphological effects and bacteriostatic properties of *N*-thiolated β-lactams suggested these compounds could potentially act on novel targets not associated with traditional antibacterial compounds. Therefore, we decided to expand testing to fungi to examine the potential for bioactivity of a selection of representative β-lactam analogs against seven *Candida* species, including *C. albicans*, *C. glabrata*, *C. tropicalis*, *C. parapsilosis*, *C. lusitaniae*, *C. kefyr*, *C. utilis* and the resilient *C. krusei*. These in vitro assays against *Candida albicans* confirmed that the lactams do indeed impede fungal cell growth [36].

Among the derivatives tested, the best antifungal compound was *N*-methythio β-lactam **1**, having a minimum inhibitory concentration of 8 µg/mL against *C. albicans*, similar to that of the clinical antifungal agent clotrimazole. The MIC value increased with time, as the incubation time increased from 24 to 48 h, indicative of static control that steadily decreases with time as its cellular concentration incrementally diminishes. The MIC values were 16 µg/mL for *Aspergillus niger* and 32 µg/mL for *Saccharomyces cerevisiae*. We noted that changing the C_2_ methoxy for an acetoxy or phenoxy decreased the bioactivity. C_2_-halogenated C_3_-aryl derivatives having fluorine attached to the aryl ring had only weak activities, while the other halogens were similarly active if they were ortho or meta to the β-lactam moiety. Having multiple halogens on the aryl ring increased the bioactivity, with, again, those halogenated at the ortho or meta centers being preferred. The most critical structural requirement, however, was found to be the nature of the organothio moiety, where *N*-methylthio was preferred over *N-sec*-butylthio (for MRSA). As in the antibacterial studies, relative and absolute stereochemistry of the β-lactam were irrelevant, as seen for enantiomeric β-lactams that gave similar growth inhibition zone sizes against all of the *Candida* species tested.

Although the concentration of β-lactam needed to induce antifungal activity far exceeds that for bacteria, the structure–bioactivity profiles that we saw were similar for the fungal and bacterial cells. Like *S. aureus*, *Candida* produces high intracellular levels of coenzyme A, and the *N*-thiolated lactams show fungistatic behavior by blocking the formation of lipids needed for cytoplasmic membranes. We examined the effect of the *N*-thiolated β-lactams on *C. albicans* cell morphology by transmission microscopy imaging and noted significant deformation of the organelles and intracellular membranes. Staining of the treated and untreated cells with trypan blue indicated that the fungal cells remained viable after treatment, indicative of the static property of the β-lactams (as for bacterial cells). The static control by the lactams enabled the cells to maturate and replicate without inducing cidal effects. None of these effects on the membranes and internal organelles were seen for the untreated cells. From these data, we postulated that the antifungal properties of these β-lactams mirror those seen against the bacterial. We confirmed that the *N*-organothio moiety of the lactams was involved in this activity by isolating cleanly the *N*-protio β-lactam after extracting the treated cells with an organic solvent. This compound had no antifungal activity and was a byproduct of the reaction within the cells. These data corroborate that it is the *N*-organothio substituent, not the β-lactam ring, that is required for antifungal activity. The subtle differences between the structure–activity preferences of the β-lactams in *S. aureus* versus *Candida*, namely the preference for an *N*-methylthio moiety for *Candida* versus an *N-sec*-butylthio for MRSA, could relate to differences in the cellular targets or transversing of the cellular membranes and cell walls.

### 2.21. Anticancer Activity of N-Thiolated β-Lactams

To further investigate the broader biological properties of these compounds, we ran them through in vitro anticancer assays, including human breast, prostate, leukemia and head-and-neck cell lines. Most of the compounds showed bioactivity, causing extensive DNA damage and the inhibition of DNA replication in Jurkat T cells [37,38]. While unexpected, this activity was the result of the activation of p38 mitogen-activated protein kinase, triggering S-phase arrest and cellular apoptosis as a result of DNA damage and leading to caspase-8 activation, the cleavage of Bcl-2 family protein Bid and the release of cytochrome C from the mitochondria. We also observed caspase-9 and caspase-3 activation along with PARP cleavage, inducing a decrease in cell membrane permeability. These events were found to be both time- and β-lactam concentration-dependent, as would be expected for caspase-induced apoptosis [39].

Similarly, we saw a time and lactam concentration dependency on DNA cleavage before arrest of the S-phase cell cycle. Comparatively, the *N*-protio lactam analog was completely inactive towards the induction of caspase-3 or PARP cleavage in the leukemia Jurkat T cells, confirming that the sulfur group is the required component to bring on cancer cell apoptosis [40]. Of the *N*-thiolated β-lactams tested, the *N*-methylthio derivative had the strongest potency compared to the *N*-ethylthio, *N*-butylthio and *N*-benzylthio analogs. This trend was also found for the rate of phosphorylation of the p32 protein upstream of caspase activation and caspase-induced apoptosis, wherein the smallest *N*-methylthio substituent was optimal. This perhaps offers a convenient way to direct the use of the different *N*-organothio β-lactams towards prokaryotic versus eukaryotic cells. Indeed, the most active compounds in in vitro antitumor experiments exert their anticancer effects at 50–100 µM, about 100–200 times higher than the bacterial MIC. At these same elevated concentrations, and even several folds higher, no cellular toxicity was observed for normal human fibroblasts, indicating significant selectivities between healthy human tissue and bacteria.

In a follow-up study, *N*-methylthio β-lactams could distinguish between cancerous versus normal cells in apoptotic effects [41]. All of the *N*-methylthio lactams inhibited colony formation of human prostate cancer cells. Most significantly, β-lactam induced apoptosis selectively in human leukemic Jurkat T and simian virus 40-transformed cells, not in non-transformed, immortalized human natural killer (NK) or normal fibroblast cells. It is likely that high glutathione levels in the redox buffer of normal mammalian cells affords protection against the apoptotic effects of the β-lactams compared to those in transformed or tumor cells [42]. These in vitro data were confirmed to extend to in vivo inhibition of breast tumor progression in mice xenografts, reducing the rate of growth of the xenograph while inducing DNA cleavage and subsequent apoptosis. Under these test conditions, we saw a 50% decrease in tumor size, with no harmful effects in the mice from treatment with the lactams.

While the cytostatic effects of these β-lactams on eukaryotic cells, and the non-cytotoxicities thus far seen in vitro, are certainly interesting and potentially exploitable, clearly, much more research is needed to gain a more complete understanding of the biological mechanisms involved in how these molecules interact with cancer versus normal healthy cells and to more confidently ascertain whether there is sufficient activity and selectivity to warrant their use in cancer chemotherapy. Until these data are obtained, caution is advised, and the current focus is thus being directed towards the antibacterial and antifungal properties that, in themselves, are highly unique and in need of much more investigation.

## 3. Conclusions and Future Prospects

The unanticipated observation in our laboratory in the mid-1990s of the antibacterial properties of *N*-methylthio-substituted β-lactams has led to an exciting exploration into the unique structural features and mechanism of action these relatively simple compounds have on antimicrobial activity. The structural features, narrow ranges of bioactivity towards bacteria, activity against eukaryotes including fungi and human cancer cell lines, their profound morphological cellular effects and ability to selectively target fatty acid biosynthesis pathways make these unique and worthy of further study. The most striking differences of these β-lactams compared to all other known β-lactams (penicillins, cephalosporins, carbapenems and monobactams) include their structural requirements (not having an ionizable side chain), their bacteriostatic properties (versus cidal for the traditional cell wall inhibitors), high stability and transparency to penicillinases and other β-lactamase proteins, stability to hydrolysis, sensitivity to thiophiles and narrow range of microbes they target. These are largely anti-MRSA agents overall, and their utility would likely be most impactful for this specific application.

In particular, their activity against highly select pathogenic microbes such as *Staphylococcus* and *Bacillus*, their unusual unique SAR patterns, visible effects on cellular components and processes completely differ to those of all other β-lactam antibiotics and antimicrobials. Another surprise is the bioactivities against eukaryotic cells, particularly yeast and cancer cells. The ongoing steady rise in antibacterial drug resistance among pathogenic strains like MRSA, and the pressing need for effective new antibiotics having alternative mechanisms of action, are important aspects to consider with these compounds. Indeed, this is where the highly narrow spectrum, bacteriostatic *N*-thiolated β-lactams could find an important niche. Moreover, their incorporation into organic nanoparticles that enhance bioactivity and perhaps bioavailability may further enable the expansion of their usefulness in the treatment of infections. Besides the compounds themselves, the ability to act selectively on alternative metabolic targets such as coenzyme A and proteins that govern metabolic processes such as lipid biosynthesis is a largely underexplored area of investigation that could provide exciting new opportunities for developing antibacterial agents against deadly pathogenic infections and the rising drug resistance. Clearly, much more research must be conducted to advance these and related molecules further towards clinical applications and to more completely evaluate their effects systemically, all requiring substantial sources of funding and sustained commitment. However, it is hoped for that the early-stage investigations summarized in this review may afford starting points and fresh insights into the drug discovery arena for infectious diseases.

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
