# Peer review of "N-Organothio β-Lactams Offer New Opportunities for Controlling Pathogenic Bacteria"

_pathogens, 2025, doi:10.3390/pathogens14070628_

Round 1

Reviewer 1 Report

Comments and Suggestions for Authors

In this review article, the author introduced N-organothio β-lactams as effective antibacterials for pathogens, especially methicillin-resistant Staphylococcus aureus (MRSA). The review covered major research efforts on histological development, stability study, structure-activity relationship, mechanistic study, and antibacterial function assays. Furthermore, the author discussed the antifungal and anticancer activities of these unique compounds.

The reviewer has a couple of minor suggestions to help improve the readability of this manuscript.

  1. Line 29-31. “A variety of methods to minimize the risk of infection by pathogenic microbes, particularly in medical centers where these microbes thrive, and interventive treatments with antibacterial agents.” This sentence has grammar issue.

Maybe change “and” to “are”. Otherwise please modify accordingly.

  1. It is suggested that the author provide more detailed background information and references in the introduction section.

For example, what are the common and widely applied antibacterials for S. aures and MRSA? What are their general principle of function towards the pathogens?

Has any resistance developed towards these antibacterials?

Based on these discussions, why are N-organothio β-lactams unique and thus draw attention from the scientific field?

  1. Please compile the chemical structures and chemical reactions into Figures with proper figure captions. The figures can be compiled by sections or by topics being discussed.

  1. It is very interesting that the N-organothio β-lactams showed selectivity on tumor tissues over normal human cells. It makes good sense that these antibacterial compounds specifically inhibit bacterial cell wall crosslinking etc. and thus have minimal toxicity to human, yet it would be better that the author discuss more on the unique effect of β-lactams on tumors, as this would draw more interested readers from the field.

Reviewer 2 Report

Comments and Suggestions for Authors

This manuscript is a high-quality review of a novel antibacterial scaffold with potential for therapeutic use beyond conventional β-lactams. Several aspects deserve special recognition:

  • The structure–activity relationship (SAR) insights are particularly valuable and will inform future design of N-organothio derivatives.

  • The discussion of glutathione and coenzyme A interactions is scientifically sound and proposes a compelling mechanistic model.

  • The review commendably integrates microbiological, biochemical, and synthetic perspectives.

Suggestions for improvement:

  1. Expand Discussion of resistance avoidance: Although resistance potential is touched upon, a dedicated section or expanded comment on how N-thiolated β-lactams may (or may not) avoid classical resistance pathways (e.g., β-lactamases, efflux, porin loss) would be beneficial.

  2. Clarify relevance of nanoparticle approach: While promising, the nanoparticle data could be better contextualized for therapeutic translation (e.g., pharmacokinetics, targeting, toxicity).

  3. Figures and diagrams: The paper would benefit from summarizing complex SAR relationships and redox interactions in visual diagrams or tables to assist readers.

  4. Antifungal and anticancer sections: While compelling, these sections could benefit from brief discussions of how these findings could be translated to in vivo or clinical applications, or their potential synergism with current treatments.

Reviewer 3 Report

Comments and Suggestions for Authors

This manuscript presents a comprehensive and in-depth analysis of a novel family of compounds—N-thiolated β-lactams—with promising antimicrobial activity. Particular emphasis is placed on their effects against Staphylococcus aureus, a clinically relevant bacterium with significant implications in public health. Moreover, the work includes an extensive structure–activity relationship (SAR) characterization, thoroughly exploring how molecular features of these compounds relate to their antimicrobial potential.

In addition to their antibacterial properties, the authors infer potential applications of these molecules as antifungal agents and anticancer compounds. This multidisciplinary scope adds significant interest and broadens the relevance of the study.

While the study is well-conceived and generally well-executed, several points merit further consideration by the authors:

For instance, in lines 663–665, the authors refer to the disruption of fatty acid biosynthesis. Although the effect is noted, the underlying molecular basis of this phenomenon is not elucidated. It should be clarified whether this effect is directly mediated by the compound’s mechanism of action or whether it results from a secondary or coadjuvant effect. In my view, this distinction should be explicitly stated in the manuscript.

Additionally, although the study provides a robust overall framework, future research could benefit from experiments involving mutant bacterial strains, particularly those with deletions or modifications in genes associated with redox metabolic pathways. This approach would provide greater mechanistic resolution regarding the proposed mode of action.

Another recommendation pertains to FabH enzyme interactions (discussed in lines 505–514). The authors might consider including molecular docking or molecular dynamics simulations in future studies to better characterize the potential binding and inhibitory effects of the compounds on this key enzyme in fatty acid metabolism.

In lines 752–756, the authors state:

    “In a follow-up study, N-methylthio β-lactams could distinguish between cancerous versus normal cells in the apoptotic effects. All of the N-methylthio lactams inhibited colony formation of human prostate cancer cells. Most significantly, β-lactam-induced apoptosis selectively in human leukemic Jurkat T and simian virus 40-transformed cells, not in non-transformed, immortalized human natural killer (NK) or normal fibroblast cells.”

While low toxicity against fibroblasts is noted, future research would benefit from in vivo bioassays, as the systemic modulation of toxicity in a whole organism is substantially different from isolated cellular models. Such results would strengthen the translational value of these findings and, in my opinion, should be included in the “Perspectives” or “Future Directions” section.

Lastly, a crucial aspect that could be further addressed in the discussion of future directions involves the adaptive potential of bacteria, fungi, and cancer cells. These organisms have evolved under fluctuating environmental conditions, often developing high phenotypic plasticity and rapid response mechanisms, which contributes to their ability to acquire drug resistance. One promising strategy to counteract this resilience could involve combinatorial treatments or the evaluation of coadjuvant effects. Such approaches may limit adaptive responses by requiring the simultaneous emergence of multiple effective mutations within a single generation. This type of strategy should be considered in future work, as it could significantly enhance the clinical efficacy and durability of therapeutic applications involving these compounds.
